# ENHANCING VISION-LANGUAGE REASONING VIA RE-INFORCEMENT LEARNING WITH SCALABLE MULTI-MODAL QA SYNTHESIS

## ABSTRACT

Building on the success of text-based reasoning models like DeepSeek-R1, extending these capabilities to multimodal reasoning holds great promise. While recent works have attempted to adapt DeepSeek-R1-style reinforcement learning (RL) training paradigms to multimodal large language models (MLLM), focusing on domain-specific tasks like math and visual perception, a critical question remains: ***How can we enhance visual-language reasoning through RL for different domains?*** To address this challenge, we make three key efforts: **(1)** A novel *Scalable Multimodal QA Synthesis* pipeline that autonomously generates domain-aware, reasoning-centric question-answer (QA) pairs directly from images across different domains. **(2)** The open-source **WeThink** dataset containing over 120K multimodal QA pairs with annotated reasoning paths, curated from 18 diverse dataset sources and covering various question domains. **(3)** A simple baseline incorporating a hybrid reward mechanism that combines rule-based verification with model-based assessment to optimize RL training efficiency across different task domains. Through comprehensive exploration of RL on our dataset, we demonstrate that the **WeThink** dataset significantly improves performance across diverse MLLM benchmarks. Furthermore, we highlight that our automated data pipeline can continuously increase data diversity, further boosting model performance.

## 1 INTRODUCTION

Visual-Language Reasoning has emerged as a pivotal capability for multimodal large language models (MLLMs), enabling tasks ranging from complex mathematical problem-solving to diverse visual question answering. Closed-source models like OpenAI's o3 (OpenAI, 2025) and Kimi k1.5 (Team et al., 2025) have demonstrated remarkable performance in visual-language reasoning, sparking significant interest within the open-source community. In contrast, recent open-source initiatives such as DeepSeek-R1 (Guo et al., 2025) have pioneered text-centric reasoning models by integrating reinforcement learning (RL) with verifiable rewards. However, these models (Chu et al., 2025; Muennighoff et al., 2025) are inherently constrained to unimodal (*i.e.*, text-only) scenarios, leaving a critical gap in multimodal reasoning capabilities.

Recent works (Chen et al., 2025a; Deng et al., 2025b; Zhang et al., 2025a; Yang et al., 2025; Wang et al., 2025b; Huang et al., 2025; Wang et al., 2025a; Wei et al., 2025; Liu et al., 2025a) have attempted to adapt DeepSeek-R1-style RL training paradigms to MLLMs, focusing primarily on domain-specific tasks like mathematical reasoning and visual perception. Yet, a key question persists: ***How can we enhance visual-language reasoning through RL for different domains?*** Two critical aspects stand out.

• **Diverse Domain-aware and Reason-centric Data**. Recent DeepSeek-R1-style methods rely on pre-collected question-answer (QA) datasets for cold-start supervised fine-tuning (SFT) with Chain-of-Thought (CoT) annotations or for reformulating answers to calculate accuracy rewards in RL. However, these QA pairs often lack the multi-step reasoning needed for robust visual-language reasoning. Additionally, some methods are dependent on domain-specific question types, which limits their scalability across various domains. To further enhance visual-language reasoning across domains, it's important to generate domains-aware and reason-focused data from diverse domains and contexts.

• **RL with Hybrid Rewards**. While rule-based rewards (e.g., answer verification for mathematical problems) are effective in specific domains, they struggle to capture the complexity of real-world multimodal scenarios, where answers can be subjective or context-dependent. This underscores the need for a hybrid reward system that combines both rule-based and model-based strategies, offering more nuanced, context-sensitive feedback to enable RL-trained MLLMs to handle diverse task domains.

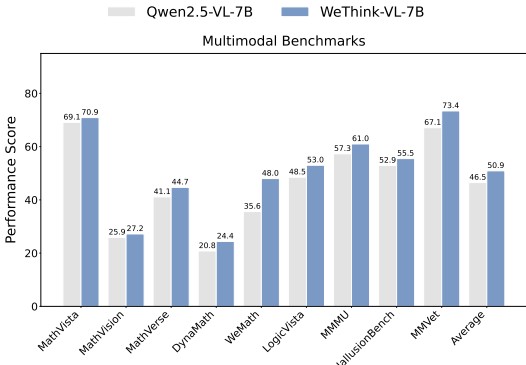

To address the data aspect, we propose an novel *Scalable Multimodal QA Synthesis* pipeline that can autonomously generate domain-aware, reason-centric questions paired with verifiable answers directly from the given images. It can benefit from diverse data sources, including

Figure 1: WeThink-VL-7B, fine-tuned on Qwen2.5-VL-7B (Bai et al., 2025) through reinforcement learning, shows significant improvements for different task domains.

open-source datasets and various resources across the Internet, enabling the continuous enhancement of data diversity. To further contribute to the field, we open-source the **WeThink** dataset, which contains over 120K multimodal QA pairs with explicit reasoning paths. Curated from 18 distinct public image datasets, **WeThink** encompasses a broad range of question domains and types, requiring integrated abilities such as *reasoning*, *OCR*, *recognition*, *math*, *knowledge*, and *spatial awareness*, thereby enhancing general multimodal reasoning capabilities.

Building upon our dataset, we establish a simple baseline that introduces a hybrid reward mechanism, integrating rule-based verification with model-based evaluation to enhance RL training efficiency across diverse task domains. Through a comprehensive exploration of RL on **WeThink**, we present four key findings: **(1)** SFT with CoT supervision on our dataset enhances the performance of less optimized model (e.g., Qwen2-VL-7B), yielding an average improvement of 3.5% across six mathematical reasoning benchmarks. **(2)** Using our dataset, direct RL fine-tuning on Qwen2.5-VL-7B (Bai et al., 2025) is sufficient and even outperforms cold-start supervised fine-tuning followed by RL. **(3)** Ablation studies on our dataset show that increasing the diversity of question domains through RL fine-tuning leads to significant improvements across tasks, from mathematical reasoning to other domains. **(4)** The scalability of our data pipeline enables continuous collection of diverse images from the Internet, further enhancing model performance, as shown in Fig. 1.

In summary, the contributions of this work are three-fold:

◇ **Automated Data Generation Pipeline**: We propose *Scalable Multimodal QA Synthesis* that autonomously generates domain-aware, reason-centric questions paired with verifiable answers directly from the given images.

◇ **Diverse Reason-centric Dataset**: We open-source the **WeThink** dataset, containing over 120K multimodal QA pairs with explicit reasoning paths, curated from distinct public datasets. It spans various question domains and types, enhancing multimodal reasoning capabilities in models.

◇ **Enhanced Visual-Language Reasoning Models**: We establish a simple baseline using a hybrid reward mechanism and conduct a comprehensive exploration of RL on our dataset, resulting in a series of models that demonstrate improved performance across diverse tasks. Furthermore, we show that the scalability of our pipeline, driven by increased data diversity, leads to further performance improvements.

## 2 RELATED WORK

### 2.1 MULTIMODAL LARGE LANGUAGE MODELS (MLLMS)

Recent years have witnessed significant advancements in Multimodal Large Language Models (MLLMs), which augment traditional Large Language Models (LLMs) by enabling them to process

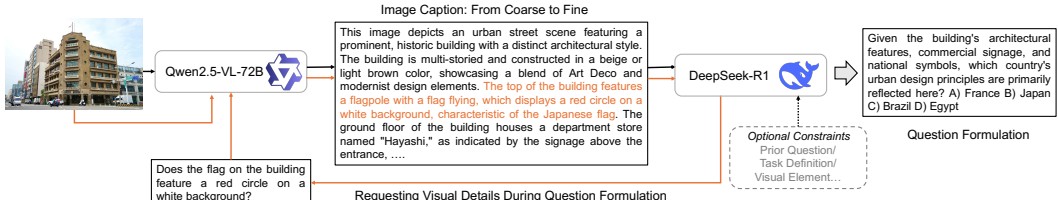

Figure 2: The automatic process of question formulation for a given image. As illustrated by the orange line, based on the coarse description provided by Qwen2.5-VL-72B, DeepSeek-R1 needs to request additional visual details (orange text) through multi-turn conversations with Qwen2.5-VL-72B, thus facilitating the generation of context-aware, reasoning-centric questions. We also highlight that the process can condition various constraints through prompts, such as prior questions (if available), task definition, and visual focus, to control the type and focus of the questions.

and comprehend information from diverse modalities, including text, images, audio, and video (Liu et al., 2023; Barrault et al., 2023; Chen et al., 2022; Li et al., 2023; Zhang et al., 2023a). The rapid evolution in this field is evidenced by the development of numerous open-source models, such as MiniGPT-4 (Zhu et al., 2023), MiniCPM-V (Yao et al., 2024), CogVLM (Wang et al., 2024c), ShareGPT4V (Chen et al., 2024a), Qwen-VL (Bai et al., 2023; Wang et al., 2024b; Bai et al., 2025), LLaVA (Liu et al., 2023; 2024b; Li et al., 2024), and InternVL (Chen et al., 2024d;c;b), alongside prominent closed-source models like Gemini (Team et al., 2023; 2024), GPT-4o (Hurst et al., 2024), Claude (Anthropic), and Grok (xAI). These efforts highlight ongoing progress in architectural designs, pre-training strategies, and instruction tuning techniques. Despite these strides, fostering robust reasoning capabilities across diverse domains and tasks continues to be a primary area of research.

## 2.2 CHAIN-OF-THOUGHT PROMPTING FOR MULTIMODAL REASONING

Chain-of-Thought (CoT) prompting, a technique that significantly enhances the reasoning capabilities of LLMs by guiding them to articulate intermediate inferential steps prior to delivering a final answer (Wei et al., 2022; Kojima et al., 2022), has been effectively extended to the multimodal domain. In MLLMs, the application of CoT not only demonstrably improves performance on complex reasoning tasks but also offers enhanced interpretability into the model's intricate decision-making processes (Zhang et al., 2023b; Lu et al., 2023b; Luo et al., 2024). A variety of strategies have been developed to elicit, generate, and leverage CoT reasoning in MLLMs. These include designing structured reasoning templates or programmatic approaches to systematically guide the CoT process (Yang et al., 2023; Zhang et al., 2024a; Mitra et al., 2024; Zheng et al., 2023; Ni et al., 2024; Chen et al., 2024e) and Supervised Fine-Tuning (SFT) using datasets enriched with multimodal CoT examples (Xu et al., 2024; Dong et al., 2024; Thawakar et al., 2025). As these strategies often generate pre-defined or limited thought processes, there is a growing focus on integrating them with reinforcement learning, to encourage exploration of diverse problem-solving strategies, and ultimately develop more sophisticated and genuinely intelligent multimodal reasoning capabilities.

## 2.3 REINFORCEMENT LEARNING FOR MULTIMODAL REASONING

Reinforcement Learning (RL) has emerged as a transformative approach for enhancing reasoning capabilities in Multimodal Large Language Models (MLLMs) (Zhou et al., 2025a). The integration of RL, particularly Reinforcement Learning from Human Feedback (RLHF) (Ouyang et al., 2022) or rule-based reward mechanisms (R1-style) (Guo et al., 2025), aims to align MLLM outputs with desired reasoning patterns and task objectives. Recently, several works have successfully adapted and extended the R1-style RL training paradigm into MLLMs. These efforts have primarily focused on exploring how R1-style RL can enhance MLLM capabilities in math-centric multi-modal reasoning (Chen et al., 2025a; Deng et al., 2025b; Zhang et al., 2025a; Yang et al., 2025; Wang et al., 2025b; Huang et al., 2025; Wang et al., 2025a; Wei et al., 2025; Liu et al., 2025a) and various specific downstream tasks. For instance, researchers have applied these RL techniques to improve scene graph understanding (Li et al., 2025; Chen et al., 2025b), visual-spatial reasoning (Zhao et al., 2025; Liao et al., 2025; Zhang et al., 2025b; Zhou et al., 2025b), referring expression comprehension (Yu et al., 2025; Shen et al., 2025; Liu et al., 2025d; Deng et al., 2025a; Liu et al., 2025b), and visual counting (Wang et al., 2025c; Tan et al., 2025; Liu et al., 2025c). While these methods have

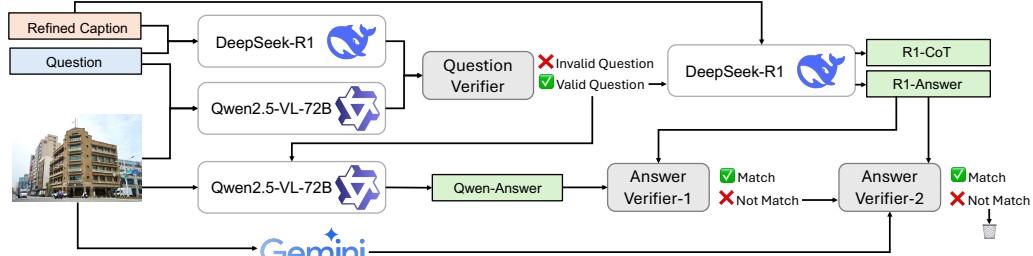

Figure 3: The automatic process of answer construction and quality control. First, DeepSeek-R1 filters out forced and open-ended questions to ensure they are verifiable. Then, using the refined caption and the valid question, DeepSeek-R1 generates a chain of thought and an answer. At the same time, Qwen2.5-VL-72B generates an answer based on the image. If their answers match, the result is kept; if not, Gemini re-evaluates DeepSeek-R1's answer, discarding incorrect responses and keeping only the correct one with its chain of thought.

demonstrated promising results within their respective scopes, few have focused on leveraging RL to broadly enhance the general multi-modal understanding and reasoning abilities of MLLMs.

## 3 WeThink Dataset with Scalable Multimodal QA Synthesis

This section presents an automated *Scalable Multimodal QA Synthesis* pipeline, designed to generate domain-aware, reasoning-centric question-answer (QA) pairs from the given images. Based on it, we introduce **WeThink**, a dataset carefully curated to encompass diverse question domains, types, and integrated abilities. Below, we describe the processes of data collection, question formulation, answer construction, and quality control, and conclude by presenting the data characteristics of **WeThink**.

### 3.1 Data Collection

Our pipeline is designed to autonomously generate high-quality QA pairs directly from the given images. These images can come from open-source datasets or various sources across the Internet. Here, to show its effectiveness, we collect open-source images to publicly release the generated QA pairs. Specifically, we sample images from 18 distinct datasets, as used in LLaVA-CoT (Xu et al., 2024). These datasets cover diverse image categories, ensuring variety and complexity in the generated QA pairs, including general images (COCO (Lin et al., 2014), SAM-1B (Kirillov et al., 2023), Visual Genome (Krishna et al., 2017), GQA (Hudson & Manning, 2019), PISC (Li et al., 2017), LLaVA (Liu et al., 2023)), text-intensive images (TextVQA (Singh et al., 2019), ShareTextVQA (Chen et al., 2024a), DocVQA (Mathew et al., 2021), OCR-VQA (Mishra et al., 2019), ChartQA (Masry et al., 2022)), scientific and technical images (GeoQA+ (Cao & Xiao, 2022), ScienceQA (Lu et al., 2022), AI2D (Kembhavi et al., 2016), CLEVR-Math (Lindström & Abraham, 2022)), and images related to art and culture contexts (WikiArt (Saleh & Elgammal, 2015; Chen et al., 2024a), Web-Landmark (Kreutzer et al., 2022; Chen et al., 2024a) Web-Celebrity (Kreutzer et al., 2022; Chen et al., 2024a)).

### 3.2 Question Formulation

Based on the collected images, we aim to generate domain-aware, reasoning-centric questions. The straightforward pipeline involves collaboration between two powerful models, Qwen2.5-VL-72B and DeepSeek-R1, to analyze images and generate questions. In this workflow, the visual-language model Qwen2.5-VL-72B first provides a detailed description of the image, after which language-only model DeepSeek-R1 analyzes the description, reflects on its content, and synthesizes relevant questions based on the analysis. However, two critical challenges arise: **(1)** incomplete visual understanding by Qwen2.5-VL-72B, **(2)** uncontrolled complexity and reasoning focus of question generation by DeepSeek-R1. To address these issues, we carefully design the question formulation process with two core strategies: *Multi-turn Information Refinements* and *Ability Synergy Constraints*, along with *Optional Contextual Constraints*.

**Multi-turn Information Refinements.** As shown in Fig. 2, given that image descriptions provided by Qwen2.5-VL-72B may sometimes be insufficient or erroneous, we implement a multi-turn information refinement mechanism to address such shortcomings, including three stages: *(1) Coarse Description Generation*: Qwen2.5-VL-72B extracts global features from the input image and generates an initial description that provides a broad overview of the main visual elements. This serves as the semantic anchor for subsequent multi-turn dialogues. *(2) Dynamic Detail Mining*: To generate context-aware, reasoning-centric questions, DeepSeek-R1 identifies information gaps based on the initial coarse description. It then generates follow-up questions to request more detailed visual information from Qwen2.5-VL-72B. This process ensures that the questions address all relevant aspects of the image, including the reasoning needed for the final question formulation. *(3) Context Integration*: As each piece of supplementary information is gathered during the multi-turn dialogue, it is integrated into the evolving description. Qwen2.5-VL-72B records this information and synthesizes it into a final, fine-grained description, which serves as the basis for generating a contextually aware question.

**Ability Synergy Constraints.** Inspired by MM-Vet benchmark (Yu et al., 2023) that evaluates the model's integrated capabilities, we propose to incorporate multi-ability constraints into the question formulation phase. This approach aims to create more complex questions that better reflect the model's ability to apply and combine the skills learned during training. Specifically, the formulation enforces mandatory reasoning capability combined with at least one complementary ability from other five-dimensional taxonomy: *a) Recognition*: General visual recognition (e.g., objects, attributes, scenes, counting, or high-level computer vision tasks); *b) Knowledge*: Use of social/visual commonsense, encyclopedic knowledge, or contextual information; *c) OCR*: Reading and reasoning over visible text (e.g., scene text, handwritten text, or embedded text in objects); *d) Spatial Awareness*: Understanding spatial relationships (e.g., object positions, directional/distance logic, layout analysis); *e) Math*: Performing arithmetic operations, solving equations, or interpreting math-specific notation. Overall, the above mechanism ensures questions inherently require: *(1) Cross-modal Reasoning Chains*: Minimum two explicit reasoning chains with comprehensive image analysis *(2) Semantics-Driven Ability Selection*: Automated activation of relevant abilities based on image content information. For instance, a question requiring object recognition + historical knowledge might ask: "Given the architectural style of the building's columns shown, what historical period does this structure represent?" This combines visual feature extraction with architectural history knowledge.

**Optional Contextual Constraints.** To achieve precise control over the generated questions, particularly in terms of their type and focus, we can optionally condition the generation process with contextual constraints. Fortunately, open-source collections often include QA pairs, and web images typically come with textual descriptions or captions. For instance, these may include prior questions to guide related queries, a clear task definition to direct the goal, and visual cues to highlight specific areas of the image. By leveraging these optional constraints, the pipeline can generate more targeted, relevant, and user-intended questions in a controlled manner.

## 3.3 Answer Construction and Quality Control

For the generated questions, the process of answers construction follows a structured approach consisting of three key stages: *Preliminary Question Filtering*, *Answer Construction and Quality Control*, as well as *CoT Refinement*. As shown in Fig. 3, each stage is carefully designed to ensure the accuracy and reliability of the answers through a multi-model verification framework.

**Preliminary Question Filtering.** Considering the instability of question formulation, our first step is to filter out questions that are unverifiable, ambiguous, or irrelevant to the image. In practice, we perform two rounds of verification. Firstly, DeepSeek-R1 uses a refined image caption to filter out invalid questions. Then, we also apply the visual-language model Qwen2.5-VL-72B to analyze the image further and filter out additional invalid questions. The remaining questions are categorized into three types: multiple-choice (MC), fill-in-the-blank (FIB), and descriptive (DES).

**Answer Construction and Quality Control.** The next stage is to generate and verify answers across different question types. For *MC* and *FIB* questions, which can be verified using rules, DeepSeek-R1 generates answers based on a refined image description, while Qwen2.5-VL-72B generates answers from the image content. These answers are then compared for alignment, and if they match, they are considered reliable. In cases of discrepancies, a secondary evaluation by another powerful visual language model Gemini 2.5 Pro (Team et al., 2024) is performed to re-evaluate and discard

incorrect answers. For **DES** questions, which often require longer, more detailed answers to stimulate reasoning and interpretation of complex visual data, DeepSeek-R1 generates the final answer, and Qwen2.5-VL-72B directly verifies its correctness. If the answer is confirmed, it is retained; otherwise, Gemini re-assesses and filters out incorrect QA pairs.

**CoT Refinement.** During the answer construction process, DeepSeek-R1 naturally generates a chain-of-thought (CoT) for each question. However, we observed that these CoTs are often overly lengthy and contain redundancies. Our subsequent experiments also revealed that these CoTs are suboptimal for both direct SFT and as cold start data for RL training. To address this, we refine the CoTs by incorporating both the image and the final answer into the QwenVL2.5-72B. This refinement process yields more concise CoTs, allowing us to more effectively investigate how CoT quality influences both SFT and its role as cold-start data for RL training.

### 3.4 DATA CHARACTERISTICS

Based on the above processes, we have constructed a new dataset **WeThink** from open-source images, which offers over **120K** comprehensive multimodal question-answer pairs with explicit reasoning paths. As a diverse and scalable resource, **WeThink** was carefully curated to encompass a broad range of question domains, types, and required integrated abilities. To better understand the dataset's structure and focus, we will analyze the following two critical aspects, with the charts presented in the appendix.

**Question Distribution.** We use Qwen2.5–VL-72B to categorize each question into five groups: general, math, chart/table/doc, knowledge, and OCR. These categories are fairly balanced. The questions are also divided into three types: multiple-choice, fill-in-the-blank, and descriptive. The first two types are suitable for RL training with rule-based rewards, while the third is used for RL training with model-based rewards. Each question type is designed for different scenarios and includes reasoning paths.

**Required Ability Distribution.** The questions in the **WeThink** dataset are designed to integrate multiple abilities, thereby controlling the difficulty level and stimulating training across various model capabilities. The core ability is reasoning, and other abilities are also triggered depending on the semantic contents of the images. Additionally, each sample engages at least two abilities simultaneously. We also howcase top-15 ability combinations in the appendix. Notably, these combinations follow a long-tail distribution, with some ability combinations being rarer than others.

## 4 A SIMPLE BASELINE WITH HYBRID REWARD

### 4.1 PROBLEM DEFINITION

Given a multi-modal input consisting of a question $q$ and an image $I$, our goal is to generate the correct answer $a$ by reasoning over both the textual and visual inputs. This reasoning process is mathematically modeled as a sequential conditional probability:

$$P(a \mid q, I) = \prod_{t=1}^{T} P(a_t \mid q, I, a_{<t}),$$

where $a_t$ is the $t$-th token of the model's output, representing a reasoning step, and $a_{<t}$ is the sequence of previously generated tokens. The model is expected to produce a logically consistent reasoning chain that integrates both the question and image, using elements such as mathematical formulas, contextual clues, and visual features. These reasoning steps should progressively lead to the final, accurate answer, bridging the textual and visual inputs in a structured manner.

### 4.2 A SIMPLE FRAMEWORK FOR REINFORCEMENT LEARNING WITH HYBRID REWARD

While supervised fine–tuning with chain-of-thought prompting provides explicit step-by-step supervision, reinforcement learning offers a complementary paradigm for optimizing reasoning generation through reward signals. Building on the success of DeepSeek-R1 in text-based reasoning tasks, we establish a simple yet effective framework for visual-language models to comprehensively explore

RL on our dataset. This framework incorporates a group-relative policy optimization strategy (Shao et al., 2024) and a hybrid reward system tailored to our dataset.

**Group-Relative Policy Optimization** eliminates value function dependency through reward normalization within response groups. For each question-image pair $(q, I)$, we sample $G$ reasoning paths $\{o_1, \ldots, o_G\}$ from the current policy $\pi_\theta$. The advantage function is computed as:

$$\hat{A}_{i,t} = \frac{R_i - \mu(\{R_j\}_{j=1}^G)}{\sigma(\{R_j\}_{j=1}^G)}$$

where $\mu$ and $\sigma$ denote the group mean and standard deviation of final rewards. The objective function combines clipped policy updates with KL regularization against the reference policy $\pi_{\text{ref}}$:

$$\mathbb{E}\left[\frac{1}{G}\sum_{i=1}^{|o_i|}\sum_{t=1}^{|o_i|}\min\left(r_{i,t}(\theta)\hat{A}_{i,t}, \text{clip}\left(r_{i,t}(\theta), 1-\epsilon, 1+\epsilon\right)\hat{A}_{i,t}\right)\right.$$

$$\left. -\beta D_{\text{KL}}(\pi_\theta \parallel \pi_{\text{ref}})\right]$$

where $r_{i,t}(\theta) = \frac{\pi_\theta(o_{i,t}|q,I,o_{i,<t})}{\pi_{\text{old}}(o_{i,t}|q,I,o_{i,<t})}$ is the importance sampling ratio. This approach stabilizes training while encouraging exploration of high-reward reasoning paths.

**Hybrid Reward System** integrates accuracy and format rewards, similar to DeepSeek-R1, with accuracy reward further divided into rule-based reward and model-based reward to handle different types of answers. For example, in our **WeThink** dataset, rule-based reward is employed for multiple-choice and fill-in-the-blank questions, while model-based reward is used for descriptive questions that require longer descriptive answers.

- **Rule-Based Reward** ($R_{\text{rule}}$): For multiple-choice and fill-in-the-blank questions, we apply exact string matching between the predicted answer and the ground truth. This is done with text normalization for case and punctuation insensitivity:

$$R_{\text{rule}} = \mathbb{I}(\text{normalize}(a_{\text{pred}}) = \text{normalize}(a_{\text{true}}))$$

where $\mathbb{I}$ is an indicator function, returning 1 for a true condition (exact match) and 0 for false (no match).

- **Model-Based Reward** ($R_{\text{model}}$): For descriptive questions, we use the DeepSeek-V3 (Liu et al., 2024a) judge model to assess answer correctness, assigning rewards based on the clarity and correctness of the response:

$$R_{\text{model}} = \begin{cases} 1 & \text{Definitely correct} \\ 0.5 & \text{Ambiguous/Partially correct} \\ 0 & \text{Definitely incorrect} \end{cases}$$

- **Format Reward** ($R_{\text{format}}$): To ensure the reasoning process is structured correctly, the format reward checks whether the response includes valid thinking and answer blocks, such as <think></think> and <answer></answer>.

$$R_{\text{format}} = \mathbb{I}(\text{Valid thinking and answer blocks})$$

where $\mathbb{I}$ returns 1 for valid (correct) formatting and 0 for invalid (incorrect) formatting.

The final reward can be computed as:

$$R = \alpha_{\text{accuracy}} \cdot R_{\text{accuracy}} + \alpha_{\text{format}} \cdot R_{\text{format}}$$

where $R_{\text{accuracy}}$ is either $R_{\text{rule}}$ or $R_{\text{model}}$, depending on the type of question. The $\alpha$ coefficients control the relative importance of the accuracy and format components. We empirically set $\alpha_{\text{accuracy}}$ to 0.7 and $\alpha_{\text{format}}$ to 0.3.

## 5 EXPERIMENTS

Due to space limitations, we provide details of benchmarks, evaluations, implementations, supplementary experiments, and case studies in the appendix B and C.

| Model | MathVista | MathVision | MathVerse | DynaMath | WeMath | LogicVista | MMMU | HallusionBench | MMVet | Average |
|---|---|---|---|---|---|---|---|---|---|---|
| | | | | Open-source Visual-Language Models | | | | | | |
| LLaVA-CoT-11B | 54.8 | - | - | - | - | - | - | 47.8 | 60.3 | - |
| LLaVA-OneVision-7B | 58.6 | 18.3 | 19.3 | 9.0 | 20.9 | 33.3 | 46.8 | 47.5 | 50.6 | 33.8 |
| InternVL2-8B | 58.3 | 20.0 | 20.4 | 9.2 | 20.2 | 33.6 | 51.2 | 45.0 | 54.3 | 34.7 |
| InternVL2.5-8B | 64.5 | 17.0 | 22.8 | 9.4 | 23.5 | 36.0 | 56.2 | 49.0 | 62.8 | 37.9 |
| Qwen2-VL-7B | 61.6 | 19.2 | 25.4 | 11.0 | 22.3 | 33.3 | 53.7 | 50.4 | 61.8 | 37.6 |
| Qwen2.5-VL-7B* | 69.1 | 25.9 | 41.1 | 20.8 | 35.6 | 48.5 | 57.3 | 52.9 | 67.1 | 46.5 |
| | | | | Open-source Visual-Language Reasoning Models | | | | | | |
| R1-VL-7B | 63.5 | 24.7 | 40.0 | - | - | - | - | - | - | - |
| X-Reasoner | 69.0 | 29.6 | - | - | - | - | 56.4 | - | - | - |
| URSA-8B-PS-GRPO | 67.8 | **31.8** | 41.5 | 22.4 | 38.3 | 44.7 | 41.1 | 41.1 | 29.9 | 39.8 |
| Visual-RFT* | 61.4 | 18.9 | 24.5 | 10.5 | 24.6 | 34.0 | 50.9 | 42.3 | 61.2 | 36.5 |
| R1-Onevision-7B* | 63.1 | 22.2 | 38.2 | 18.4 | 33.2 | 44.6 | 52.1 | 49.7 | 62.5 | 42.7 |
| VLAA-Thinker-7B | 68.0 | 26.4 | **48.2** | 22.4 | 41.5 | 48.5 | 56.6 | 51.9 | 65.8 | 47.7 |
| WeThink-VL-7B | **71.6** | 26.7 | 45.1 | 24.0 | 45.5 | 51.9 | 59.3 | **55.8** | 71.7 | 50.2 |
| WeThink-VL-7B† | 70.9 | 27.2 | 44.7 | **24.4** | **48.0** | **53.0** | **61.0** | 55.5 | **73.4** | **50.9** |

Table 1: Comparison of various VLMs across diverse multimodal benchmarks. We evaluate the official model checkpoint provided, using VLMEvalKit, marked with a ∗. The best results are highlighted in **bold**, while the second-best results are underlined. A superscript † indicates the use of external images from the Internet to further enhance diversity.

| Reward Type | MathVista | MathVision | MathVerse | DynaMath | WeMath | LogicVista | Average |
|---|---|---|---|---|---|---|---|
| Qwen2.5-VL-7B* | 69.1 | 25.9 | 41.1 | 20.8 | 35.6 | 48.5 | 40.2 |
| *Rule* | 65.9 | 25.1 | 42.6 | 24.0 | 39.1 | 45.2 | 40.3 |
| *Model* | 63.0 | 24.9 | 43.3 | 25.7 | 31.9 | 45.6 | 39.1 |
| *Rule+Model* | 66.8 | 26.2 | 45.7 | 24.2 | 37.9 | 47.4 | **41.4** |

Table 2: The impact of RL with different reward types on all ***math-type*** questions in **WeThink**.

## 5.1 Implementation Details

The implementation of our simple framework is built on EasyR1 (Zheng et al., 2025), which is based on veRL (Sheng et al., 2024). We perform 5 samples per query with a temperature setting of 1.0. We select Qwen2.5-VL-7B-Instruct as our base model and perform full-parameter RL fine-tuning, with rollout and training batch sizes set to 512 and 128, respectively. For ablation studies, we use 8 NVIDIA H20s for all experiments, while 32 NVIDIA H20s is used for full-scale training on **WeThink**. The DeepSeek-V3 judge model is deployed on 16 NVIDIA H20s and participates in the RL training through API calls for reward computation.

## 5.2 Main Results

As shown in Tab. 1, WeThink-7B outperforms open-source vision-language models across various task domains, with a 4.4-point average improvement over the base model Qwen2.5-VL-7B. Compared to concurrent models, WeThink-7B shows broader improvements across different domains, not only in mathematics but also on other benchmarks. In contrast, models like R1-Onevision-7B and VLAA-Thinker-7B, which also use Qwen2.5-VL-7B, focus on domain-specific performance but suffer declines in other areas. This highlights the effectiveness of the WeThink dataset, which enhances multimodal reasoning by covering diverse domains and tasks. It also demonstrates that our pipeline brings advantages in terms of new domain data and tasks.

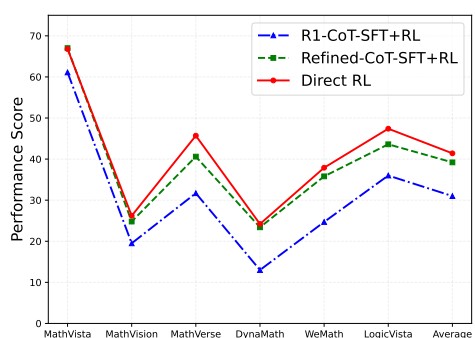

Figure 5: The impact of cold start SFT for RL on all ***math-type*** questions in **WeThink**.

## 5.3 Ablation Study

**The impact of cold start during RL.** Following DeepSeek-R1, we initialize RL training with the CoT SFT model as the cold-start checkpoint, trained on 120K WeThink samples using CoT prompting. For subsequent RL training, we select all math-type questions from WeThink and incorporate hybrid rewards. Fig. 4 demonstrates that using refined CoT data for cold start significantly enhances model

| Question Type | MathVista | MathVision | MathVerse | DynaMath | WeMath | LogicVista | MMMU | HallusionBench | MMVet | Average |
|---|---|---|---|---|---|---|---|---|---|---|
| Qwen2.5-VL-7B | 69.1 | 25.9 | 41.1 | 20.8 | 35.6 | 48.5 | 57.3 | 52.9 | 67.1 | 46.5 |
| *Math* | 66.8 | 26.2 | 45.7 | 24.2 | 37.9 | 47.4 | 57.2 | 56.0 | 72.2 | 48.2 |
| *All* | 71.6 | 26.7 | 45.1 | 24.0 | 45.5 | 51.9 | 59.3 | 55.8 | 71.7 | **50.2** |

Table 3: The impact of RL with different question types (*i.e., **math-type** and **all-type***) in **WeThink**, comparing performance across diverse multimodal benchmarks.

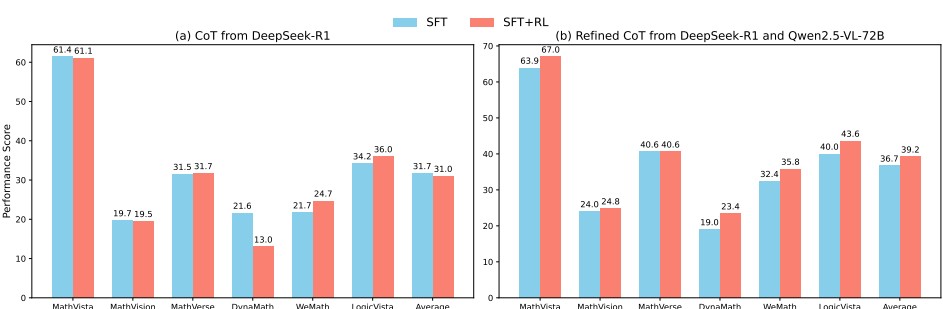

Figure 4: The impact of cold start data quality.

performance following RL. Conversely, applying RL directly to a model initialized with the original R1-CoT checkpoint yields negligible improvements. These results clearly underscore the critical importance of high-quality cold-start data for effective RL enhancement. Further reinforcing this point, Fig. 5 shows that direct RL fine-tuning on the base Qwen2.5-VL-7B model leads to better gains than the cold-start method. This suggests that when leveraging our dataset, direct RL optimization on Qwen2.5-VL-7B is a highly effective strategy.

**The impact of RL with different reward types.** We divide questions in WeThink into rule-based reward questions (*i.e.*, multiple-choice and fill-in-the-blank questions) and model-based reward questions (*i.e.* descriptive questions), and train each model for the same steps. Our results in Tab. 2 show that hybrid rewards yield the best performance, with an average improvement of 1.2% across six benchmarks.

**The impact of RL with different question types.** To further enhance visual-language reasoning across domains beyond mathematics, we extend RL training to include all question types in the WeThink dataset. As shown in Tab. 3, we compare models trained exclusively on math-type questions with those trained on the full range of question types, evaluated across diverse benchmarks. The results show that using all question types not only boosts performance on math benchmarks but also improves results on general tasks.

**The impact of RL with increasing data diversity.** To demonstrate the scalability of our data generation pipeline, we collect approximately 20K in-the-wild images from the Internet. Following the same data construction process, we generate new QAs for these images and incorporate them into the training. The results presented in Tab. 1 show that performance continues to improve across diverse benchmarks. However, the addition of new data leads to a slight decline in some benchmarks, such as MathVista and MathVerse. This suggests that RL training process is sensitive to changes in the data distribution for specific benchmarks. Nonetheless, the average improvements show our pipeline's scalability in incorporating more data to further enhance performance.

## 6 CONCLUSION

This work advances multimodal understanding by enhancing visual-language reasoning through reinforcement learning (RL). We introduce a novel *Scalable Multimodal QA Synthesis* pipeline to generate domain-aware, reasoning-focused question-answer pairs from images across different domains. We also release the **WeThink** dataset, with over 120K multimodal QA pairs and annotated reasoning chains, to improve RL training across domains. We establish a simple baseline and conduct comprehensive exploration of RL on our dataset, incorporating a hybrid reward mechanism. Results across various MLLM benchmarks show the dataset's effectiveness in improving task performance, while our automated pipeline ensures continuous data diversity and scalable RL training.

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

| Image Type | Source Dataset | Images |
|---|---|---|
| General Images | COCO | 30786 |
| | SAM-1B | 12014 |
| | Visual Genome | 4414 |
| | GQA | 3483 |
| | PISC | 1148 |
| | LLaVA | 150 |
| Text-Intensive Images | TextVQA | 17571 |
| | ShareTextVQA | 429 |
| | DocVQA | 5805 |
| | OCR-VQA | 6485 |
| | ChartQA | 22865 |
| Scientific & Technical | GeoQA+ | 4607 |
| | ScienceQA | 3236 |
| | AI2D | 12024 |
| | CLEVR-Math | 434 |
| Art & Culture | WikiArt | 401 |
| | Web-Landmark | 256 |
| | Web-Celebrity | 319 |

Table 4: The distribution analysis of image types from **WeThink**.

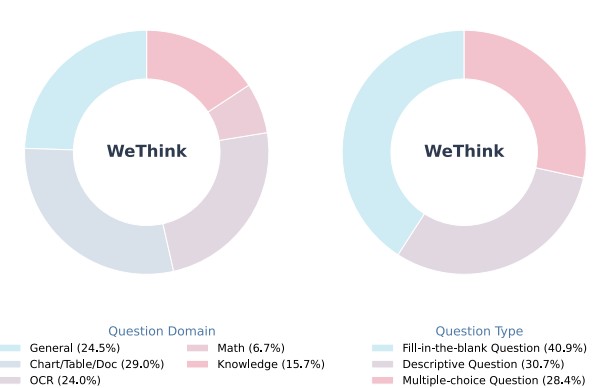

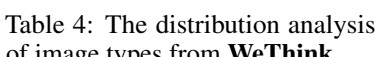

Figure 6: The distribution analysis of question domains and types from **WeThink**.

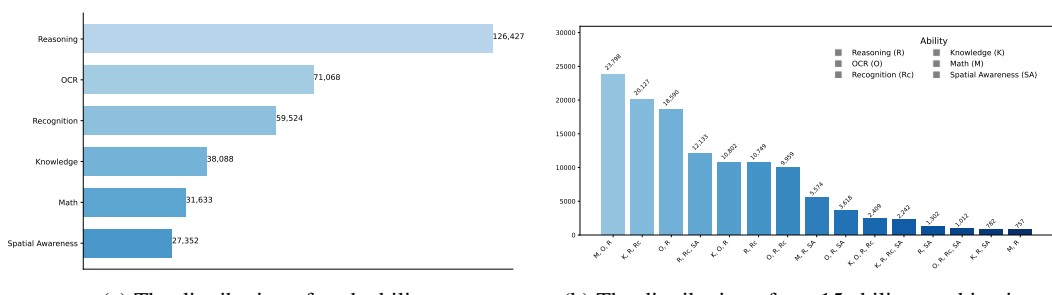

(a) The distribution of each ability      (b) The distribution of top-15 ability combinations

Figure 7: We show the difficutity distribution of question sample in the **WeThink** dataset.

## A  THE DETAILS OF SCALABLE MULTIMODAL QA SYNTHESIS

We formulate the question formulation designs into a prompt protocol for DeepSeek-R1, which is structured into five parts:

**Input Information:** The input consists of potentially insufficient or erroneous preliminary visual information from Qwen2.5-VL-72B. DeepSeek-R1 relies on Qwen2.5-VL-72B to continuously refine this visual information and use it as updated input for generating reasoning-based questions through a multi-turn process.

**Core Requirements**: For each generated question, it is essential that multiple abilities are triggered, ensuring the depth and complexity of reasoning. Typically, this involves at least two reasoning steps, making the question both logical and comprehensive. Furthermore, the question design must rely on thorough image analysis to maintain clarity, depth, and completeness.

**Question Designs**: To ensure high-quality output, we have established specific guidelines for the selection of abilities. Every question must include reasoning as a mandatory component. In addition, depending on the image content, the question should also incorporate at least one of the following abilities: recognition (feature extraction), knowledge (external knowledge application), OCR (text recognition), spatial awareness (geometric or positional reasoning), or math (numerical reasoning). Moreover, we can optionally condition the generation process with contextual constraints, to achieve better control questions' type and focus.

**Information Request Protocol**: In practice, when the image description provided by Qwen2.5-VL-72B is insufficient, we activate the Information Request Protocol, which allows for up to three rounds of clarification requests to ensure that the necessary visual information is complete for generating the subsequent questions.

**Output Specifications**: Finally, the output of the question generation process must adhere to strict format specifications, including clarification requests for insufficient information (e.g., `<clarify>...</clarify>`) and properly formatted valid questions (e.g., `<q>...</q>`).

## B  EXPERIMENTAL IMPLEMENTATION DETAILS

The anonymous code and dataset are available at `https://anonymous.4open.science/r/WeThink-7C9A` and `https://huggingface.co/datasets/WeThink/WeThink-Multimodal-Reasoning-120K`.

### B.1  BENCHMARKS & EVALUATION

To comprehensively evaluate multi-modal understanding and reasoning capabilities of our models, we conduct experiments across diverse benchmarks, including MathVista (Lu et al., 2023a), MathVision (Wang et al., 2024a), MathVerse (Zhang et al., 2024b), DynaMath (Zou et al., 2024), We-Math (Qiao et al., 2024), and LogicVista (Xiao et al., 2024), MMMU (Yue et al., 2024), MMVet (Yu et al., 2023), HallusionBench (Guan et al., 2024). The above benchmarks are available on the Open-Compass MLLM Leaderboard. To maintain fairness and reproducibility, we evaluate our models using VLMEvalKit (Duan et al., 2024), which is an open-source toolkit for MLLM evaluation.

### B.2  SUPERVISED FINE-TUNING USING WETHINK

**Implementation Details.** We conduct chain-of-thought prompting using over 120K CoT-annotated diverse QA pairs from our **WeThink** dataset for direct SFT. We investigate the impact of two types of CoT (*i.e.*, the original R1-CoT and the refined CoT) on powerful open-source models. In practice, we perform full-parameter fine-tuning for 1 epoch using 8 NVIDIA H20 GPUs on two instruction-tuned models: Qwen2-VL-7B and Qwen2.5-VL-7B.

**Main Results.** As shown in Tab. 5, the experimental results reveal two key findings: (1) ***The quality of the SFT CoT is critical.*** The original R1-CoT is overly long and redundant, as it only directs the model to mimic the reasoning templates in the annotated CoT structure. Even with less optimized models like Qwen2-VL-7B, fine-tuning leads to improvements on certain benchmarks (e.g., a 5% improvement on DynaMath). However, for the more advanced Qwen2.5-VL-7B model, we observe significant performance degradation across all benchmarks. (2) ***Our CoT data is particularly beneficial for less optimized models.*** Specifically, for Qwen2-VL-7B, the fine-tuning results in an average improvement of 3.5%. In contrast, applying direct SFT to the well-optimized Qwen2.5-VL-7B leads to a drop in performance.

### B.3  SYSTEM PROMPT

To structure the reasoning process during training, we use the following system prompt for both supervised fine-tuning and reinforcement fine-tuning settings as follow:

---
**System Prompt**

```
"You FIRST think about the reasoning process as an internal
monologue and then provide the final answer.  The reasoning
process MUST BE enclosed within <think></think>tags.  The
final answer MUST BE enclosed within <answer></answer>tags."
```
---

### B.4  BENCHMARKS & EVALUATION.

As illustrated in Sec. B.1, we conduct comprehensive evaluation across 14 MLLM benchmarks, covering six mathematical reasoning benchmarks and eight general multimodal understanding benchmarks. Below are the details:

| Model | MathVista | MathVision | MathVerse | DynaMath | WeMath | LogicVista | Average |
|---|---|---|---|---|---|---|---|
| Qwen2-VL-7B* | 61.8 | 19.0 | 25.6 | 11.0 | 21.4 | 34.7 | 28.9 |
| R1-CoT SFT | 56.6 | 17.0 | 25.0 | 16.0 | 18.5 | 35.8 | 28.2 |
| Refined-CoT SFT | 59.5 | 20.3 | 32.9 | 15.2 | 25.0 | 41.2 | **32.4** |
| Qwen2.5-VL-7B* | 69.1 | 25.9 | 41.1 | 20.8 | 35.6 | 48.5 | **40.2** |
| R1-CoT SFT | 61.4 | 19.7 | 31.5 | 21.6 | 21.7 | 34.2 | 31.7 |
| Refined-CoT SFT | 63.9 | 24.0 | 40.6 | 19.0 | 32.4 | 40.0 | 36.7 |

Table 5: The impact of fully supervised fine-tuning on **WeThink**. * denotes the model results reproduced by us. We highlight the best average results in **bold**.

**Mathematical reasoning** employs the following benchmarks: ⋄ MathVista_MINI, which is the Test Mini split of the MathVista dataset

⋄ MathVision, which uses the full test set of MathVision

⋄ MathVerse_MINI_Vision_Only, the Test Mini split of MathVerse, using the "Vision Only" mode

⋄ DynaMath, which uses the full test set of DynaMath

⋄ WeMath, the Test Mini split of WeMath, where we report "Score (Strict)" as the main metric

⋄ LogicVista, which is the full test set of LogicVista

**General multimodal understanding** employs MMBench v1.1 (Test_CN / Test_EN), MMStar, MMMU (Val), HallusionBench, AI2D (Test), OCRBench, MMVet and RealWorldQA.

**Evaluation Details.** To ensure fairness and reproducibility, we conduct evaluation of our models using VLMEvalKit Duan et al. (2024), an open-source toolkit designed for MLLM evaluation. All evaluations are conducted using 8 NVIDIA A800s. We follow the Qwen2.5-VL series settings, where the minimum number of pixels is set to 1280*28*28, the maximum number of pixels is set to 16384*28*28, and the `use_custom_prompt` option is disabled. The `max_new_tokens` parameter is set to 2048 by default, in alignment with Qwen2.5-VL series. The system prompt is provided in Tab. B.3. We empirically found that using the default system prompt of the Qwen2.5-VL series yields better performance for OCRBench and AI2D benchmarks.

## C  CASE STUDY: FROM MATH TO GENERAL CHALLENGES

Below, we conduct case study to investigate how reasoning contributes to performance improvements in diverse scenarios including *Mathematics*, *Medicine*, *Humanities*, *Science*, *Art*, *Engineering* and *Daily Scenes*. These examples show that WeThink-VL-7B exhibits superior logical and analytical capabilities, leading to correct answer.

---

**Mathematics**

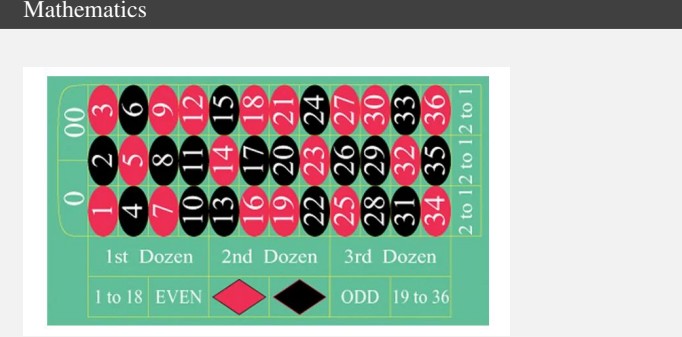

Question: The casino game, roulette, allows the gambler to bet on the probability of a ball, which spins in the roulette wheel, landing on a particular color, number, or range of numbers. The table used to place bets contains of 38 numbers, and each number is assigned to a color and a range. Compute the probability of winning the following type of bets: Betting on three numbers in a line, as in 1-2-3
Options:
A. 1/38
B. 2/38

---

C. 3/38
D. 6/38
GT Answer: C

---

**Qwen2.5-VL-7B**

To determine the probability of winning a bet on three numbers in a line (such as 1-2-3) in roulette, we need to understand the total number of possible outcomes and the number of favorable outcomes.
1. Total number of possible outcomes: There are 38 numbers on the roulette wheel.
2. Favorable outcomes: There is only 1 way to win by betting on the exact three numbers 1-2-3.
The probability $P$ of winning is given by the ratio of the number of favorable outcomes to the total number of possible outcomes:

$$P = \frac{\text{Number of favorable outcomes}}{\text{Total number of possible outcomes}} = \frac{1}{38}$$

Therefore, the correct answer is:
A. 1/38

---

**WeThink-VL-7B:**

<think>To compute the probability of winning by betting on three numbers in a line, we need to understand that there are 38 possible numbers on the roulette wheel. When you bet on three specific numbers, you are essentially looking for the probability that the ball lands on one of those three numbers. The probability of the ball landing on any specific number is 1/38. Since you are betting on three specific numbers, the probability of winning is the sum of the probabilities of the ball landing on each of those three numbers. However, since these are mutually exclusive events (the ball can't land on more than one number at a time), we simply add the probabilities:

$$\text{Probability} = \frac{1}{38} + \frac{1}{38} + \frac{1}{38} = \frac{3}{38}$$

So, the probability of winning by betting on three numbers in a line is 3/38. </think>
<answer>C. 3/38 </answer>

---

Medicine

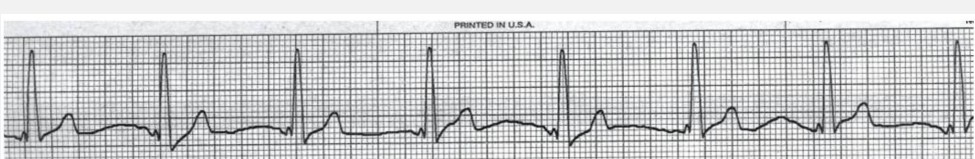

Question: What is this rhythm?
Options:
A. Normal Sinus Rhythm
B. Sinus Bradycardia
C. Accelerated Junctional Rhythm
D. NSR with 1st Degree AV Block
E. Idioventricular Rhythm
F. Paced with Failure to Capture
GT Answer: C

---

**Qwen2.5-VL-7B**

The rhythm in the provided ECG appears to be irregular with a slow heart rate. The P waves are not clearly visible, and the QRS complexes are wide and irregular. This suggests a rhythm that is not sinus rhythm. Given the options provided, the most likely diagnosis is:
E. Idioventricular Rhythm
Idioventricular rhythm is characterized by wide, irregular QRS complexes and a slow heart rate, which matches the appearance of the rhythm in the ECG.

---

**WeThink-VL-7B:**

<thinkimage provided is an ECG (Electrocardiogram) which is used to monitor the electrical activity of the heart. The rhythm in the ECG shows regular, narrow QRS complexes with no P waves preceding

them, which is characteristic of junctional rhythms. The QRS complexes are regular and the heart rate appears to be relatively slow, which is not typical for normal sinus rhythm or bradycardia. The absence of P waves and the regularity of the QRS complexes suggest a junctional rhythm. The regularity and the narrow QRS complexes also rule out idioventricular rhythm, which is typically irregular and wide QRS complexes. The paced rhythm with failure to capture is not applicable as there are no artificial pacing spikes visible. The rhythm is not consistent with first-degree AV block as the QRS complexes are narrow and regular. Therefore, the most likely rhythm is an accelerated junctional rhythm. </think>
<answer>C. Accelerated Junctional Rhythm </answer>

## Humanities

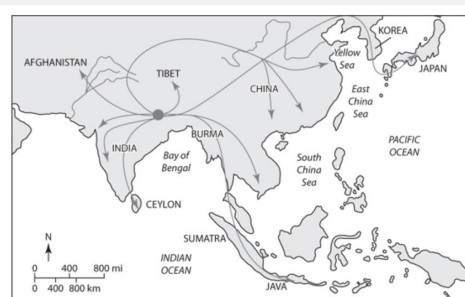

Question: A historian would use the above map to illustrate which of the following?
Options:
A. Major trade routes in the classical era
B. The spread of communism from 1900 to today
C. One aspect of cultural diffusion
D. The monsoonal wind patterns of Asia
GT Answer: C

---

**Qwen2.5-VL-7B**
The correct answer is:
A. Major trade routes in the classical era
The map shows various routes connecting different regions, which is indicative of trade and cultural exchange patterns. The presence of arrows and the geographical context suggest the movement of goods and ideas, which is characteristic of major trade routes in the classical era.

---

**WeThink-VL-7B:**
<think>The map shows various routes connecting different parts of Asia, including China, India, and the islands of Southeast Asia. These routes are likely to represent trade or cultural exchanges rather than wind patterns or the spread of communism. The presence of arrows and the interconnected nature of the lines suggest movement and exchange, which is characteristic of trade routes or cultural diffusion. The map does not specifically indicate monsoonal wind patterns, as it lacks the necessary meteorological data and is more focused on geographical and cultural connections. </think>
<answer>C. One aspect of cultural diffusion </answer>

## Science

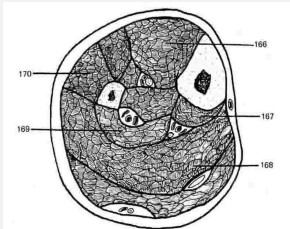

Question: In the Section of left leg, identify the 170 structure.

Options:
A. Tibialis anterior
B. Tibialis posterior
C. Flexor hallucis longus
D. Peroneus longus
GT Answer: D

---

**Qwen2.5-VL-7B**
The structure labeled as 170 in the image appears to be the tibialis posterior muscle, which is located in the posterior compartment of the leg. Therefore, the correct answer is:
B. Tibialis posterior

---

**WeThink-VL-7B:**
<think>The image provided is a cross-sectional view of a leg, likely representing the anatomy of the lower leg. The structure labeled as 170 appears to be located on the outer side of the section, which corresponds to the lateral side of the leg. In the anatomy of the leg, the tibialis posterior is typically located on the medial side, the tibialis anterior on the anterior side, the flexor hallucis longus is deep within the leg, and the peroneus longus is on the lateral side, which matches the location of the structure labeled 170. </think>
<answer>The correct answer is D. Peroneus longus. </answer>

---

**Art**

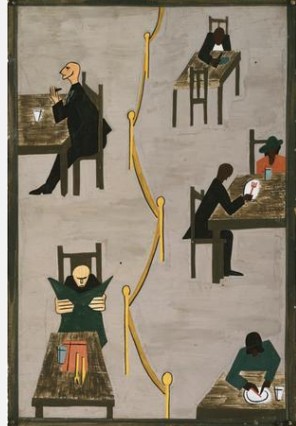

Question: The painting shown was created by:
Options:
A. Wifredo Lam
B. Frida Kahlo
C. Diego Rivera
D. Jacob Lawrence
GT Answer: D

---

**Qwen2.5-VL-7B:**
The painting shown is "The Four Stages of a Man" by Wifredo Lam. Therefore, the correct answer is:
A. Wifredo Lam

---

**WeThink-VL-7B:**
<think>The image depicts a painting with a series of figures engaged in various activities, such as reading, writing, and working. The style of the painting, with its flat, almost cut-out appearance of the figures and the use of a single, prominent line that seems to connect the different scenes, is characteristic of the work of Jacob Lawrence. Jacob Lawrence is known for his series of paintings that often depict historical events or social issues, and his style often includes a similar use of bold lines and a focus on the human figure. The other artists listed have distinct styles: Wifredo Lam is known for his surrealist works, Frida Kahlo for her self-portraits and symbolic works, and Diego Rivera for his large-scale murals and political themes. The style of this painting does not align with the known styles of the other artists. </think>

<answer>D. Jacob Lawrence </answer>

---

## Engineering

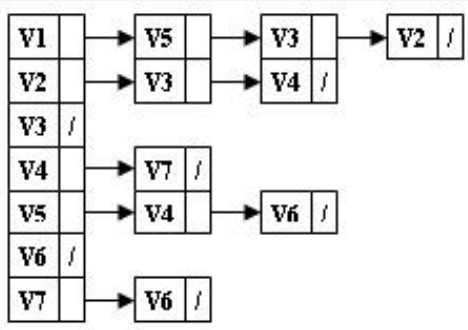

Question: The adjacency list of a directed graph is given as follows. Starting from the vertex V1 and traversing according to the depth-first search method, the obtained vertex sequence is:
Options:
A. V1,V5,V4,V7,V6,V2,V3
B. V1,V2,V3,V4,V7,V6,V5
C. V1,V5,V4,V7,V6,V3,V2
D. V1,V5,V6,V4,V7,V2,V3
GT Answer: C

---

**Qwen2.5-VL-7B:**
To determine the correct vertex sequence obtained by traversing the graph using a depth-first search (DFS) starting from vertex V1, we need to follow the adjacency list and the DFS rules:
1. Start at V1.
2. Visit V1 and then follow the edges to its neighbors.
3. Continue this process, visiting each vertex and its neighbors, ensuring that each vertex is visited only once.
Let's follow the adjacency list step-by-step:
1. Start at V1.
2. V1 points to V5, so visit V5.
3. V5 points to V4, so visit V4.
4. V4 points to V7, so visit V7.
5. V7 points to V6, so visit V6.
6. V6 has no further neighbors, so backtrack to V5.
7. Since V5 has already been visited, backtrack to V4.
8. Since V4 has already been visited, backtrack to V1.
9. Since V1 has already been visited, backtrack to V2.
10. V2 has no further neighbors, so backtrack to V3.
11. V3 has no further neighbors, so backtrack to V1.
The sequence of vertices visited is: V1, V5, V4, V7, V6, V2, V3.
Thus, the correct answer is: A. V1,V5,V4,V7,V6,V2,V3

---

**WeThink-VL-7B:**
<think>To determine the correct vertex sequence obtained by traversing the graph from vertex V1 using a depth-first search (DFS) method, we need to follow the adjacency list and the rules of DFS. DFS typically involves visiting a vertex, then exploring as far as possible along each branch before backtracking.
Starting from V1:
1. V1 ->V5 (first child)
2. V5 ->V4 (first child)
3. V4 ->V7 (first child)
4. V7 ->V6 (first child)
5. V6 ->(no further children, backtrack)
6. V4 ->(already visited, backtrack)
7. V5 ->V3 (next child)

8. V3 ->V2 (first child)
9. V2 ->(no further children, backtrack)
10. V3 ->(already visited, backtrack)
11. V5 ->(already visited, backtrack)
12. V1 ->(already visited, backtrack)
The sequence obtained is: V1, V5, V4, V7, V6, V3, V2.
Let's match this with the given options:
A. V1,V5,V4,V7,V6,V2,V3
B. V1,V2,V3,V4,V7,V6,V5
C. V1,V5,V4,V7,V6,V3,V2
D. V1,V5,V6,V4,V7,V2,V3
The correct sequence is: V1, V5, V4, V7, V6, V3, V2, which matches option C. </think>
<answer>C. V1,V5,V4,V7,V6,V3,V2 </answer>

### Daily Scenes

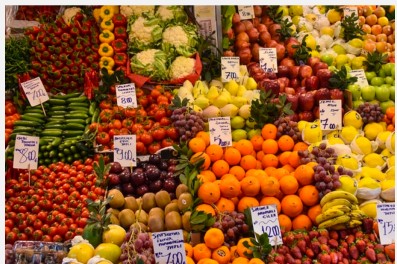

Question: What fruit is the right of plums?
GT Answer: orange

---

**Qwen2.5-VL-7B**

The fruit to the right of the plums in the image is a kiwi.

---

**WeThink-VL-7B:**

<think>The image shows a variety of fruits and vegetables. The plums are located in the middle of the image, slightly to the left. To the right of the plums, there are several oranges. The oranges are clearly visible and are positioned next to the plums. </think>
<answer>The fruit to the right of the plums is an orange. </answer>

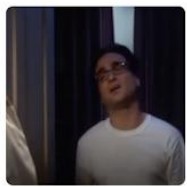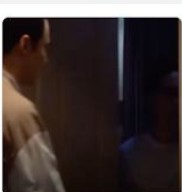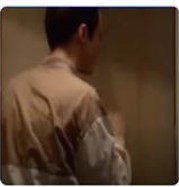

Question: According to the positive sequence of the images, does the man close the door?
GT Answer: Yes

---

**Qwen2.5-VL-7B**

No, the man does not close the door in the sequence of images provided. The images show the man standing in front of a door, but there is no indication that he closes the door. The sequence appears to show the man facing the door, possibly preparing to open it, but the door is not closed in the images.

---

**WeThink-VL-7B:**

<think>The sequence of images shows a man in a white shirt. In the first image, he is facing someone or something off-camera. In the second image, he is seen from the side, and it appears he is moving towards a door. In the third image, he is seen from behind, and it looks like he is interacting with the door, possibly closing it. The progression of the images suggests a movement towards the door and a possible action of closing it. </think>
<answer>Yes, according to the positive sequence of the images, the man appears to close the door. </answer>

