# OpenReview forum: "Enhancing Vision-Language Reasoning via Reinforcement Learning with Scalable Multimodal QA Synthesis"
_ICLR.cc/2026/Conference — ICLR 2026 Conference Withdrawn Submission_

### Official Review · Reviewer_dNe2 · 2025-10-25

**Soundness:** 3
**Presentation:** 3
**Contribution:** 2
**Rating:** 4
**Confidence:** 4

**Summary:**

The paper addresses the problem of “how to enhance multimodal reasoning across domains via reinforcement learning (RL)” and proposes three main contributions:
(1) a scalable multimodal QA automatic synthesis pipeline that generates domain-aware, reasoning-oriented, and verifiable QA pairs from images;
(2) an open dataset, WeThink, containing over 120K samples with explicit reasoning chains, sourced from 18 public image datasets;
(3) a concise RL baseline based on hybrid rewards (rule-based verification + model-based judgment), with reported improvements and ablations on multiple VLM benchmarks (including comparisons between math-only and all-question-type training, and between different reward schemes).
The key idea of the pipeline is as follows: Qwen2.5-VL-72B first provides coarse-to-fine image descriptions, then DeepSeek-R1 performs multi-round refinement and applies a “capability coordination constraint” to ensure questions require cross-capability reasoning and verifiability. High-quality samples are then constructed via multi-model validation and CoT-based refinement.
For evaluation, the authors employ VLMEvalKit to align testing across several mathematical and general multimodal benchmarks, demonstrating that WeThink-VL-7B outperforms its base model Qwen2.5-VL-7B on average, and that hybrid rewards yield a 1.2% average improvement over single-reward settings.

**Strengths:**

1. The paper systematically transfers the DeepSeek-R1-style verifiable RL paradigm to the multimodal setting and proposes a problem-generation framework combining “capability coordination constraints + multi-round refinement.” Compared with direct question generation from descriptions, this dual-model adversarial design is more targeted and emphasizes the verifiability of the data, showing methodological novelty.

2. The dataset construction includes multi-level quality control: pre-filtering of problematic questions, dual-model answer comparison, third-model verification, and subsequent CoT-based refinement to reduce redundancy. The overall process is well-structured and likely reduces the introduction of unverifiable, ambiguous, or incorrect QA pairs.

3. WeThink covers multiple domains and question types with explicit reasoning chains. If released, it could serve as a valuable cold-start data source for multimodal reasoning in RL/SFT settings. Under equivalent training resources, WeThink-VL-7B outperforms its base model and achieves consistent improvements across benchmarks, indicating strong practical value.

**Weaknesses:**

1. During quality control, samples are retained if DeepSeek-R1 and Qwen2.5-VL-72B provide consistent answers, and otherwise re-evaluated by Gemini. This correctness criterion depends on model agreement rather than independent annotation, which may amplify systematic bias among models. The paper does not report the error rates, rejection rates, or human-audited comparisons of these consistency and verification steps, which limits confidence in data reliability and reusability.

2. The pipeline and training/evaluation process rely on closed-source or large-model APIs (e.g., Gemini, DeepSeek-V3 judge). While feasible from an engineering standpoint, such dependence undermines reproducibility and accessibility for the academic community. It is recommended to disclose open-source alternatives or degraded variants and compare their performance.

3. The images in WeThink are mainly sourced from 18 public datasets, with an additional ~20,000 images collected from the internet. The authors acknowledge slight performance degradation on certain math benchmarks, suggesting sensitivity to distributional shift. However, the paper does not report overlap or near-duplicate detection between WeThink and evaluation benchmarks, raising potential concerns about information leakage or data contamination.

4. Although the paper reports a 1.2% average improvement of hybrid reward over single reward, it lacks detailed analysis of the model discriminator’s scoring scheme (e.g., decision thresholds, scale stability, sensitivity to question type/length) and the robustness of the rule-based reward (case sensitivity, unit handling, synonymous expressions). Without such decomposition and error analysis, it is difficult to pinpoint the exact sources of gain or failure patterns.

5. The authors note that direct SFT on a strong base model leads to degraded performance, and that “direct RL outperforms SFT+RL cold-start.” However, learning curves, KL constraints, sampling temperature/group size, and other critical hyperparameter comparisons or statistical significance tests are not provided, making it difficult to reproduce the claimed superiority of direct RL.

6. The “capability coordination constraint” in question generation is only qualitatively described. There is no disclosure of executable templates or rule libraries for triggering different semantic capabilities, nor examples of failure cases. Providing such templates would significantly enhance the reproducibility and usability of the approach.

**Questions:**

1. Please provide the consistency rate between DeepSeek-R1 and Qwen2.5-VL-72B answers, their agreement rate with human annotations, and the proportion of samples judged incorrect/discarded by Gemini. In addition, include a stratified confusion matrix (by question type/domain) and error analysis to support conclusions about data reliability.

2. The issues described in the “Weaknesses” section.

---

### Official Review · Reviewer_eVp6 · 2025-10-30

**Soundness:** 3
**Presentation:** 3
**Contribution:** 2
**Rating:** 4
**Confidence:** 5

**Summary:**

The paper presents a scalable multimodal question-answering (QA) synthesis pipeline designed to automatically generate domain-aware, reasoning-centric QA pairs. Building upon this pipeline, the authors construct the WeThink dataset, which comprises over 120K multimodal QA pairs accompanied by explicit reasoning paths. Furthermore, the paper introduces a hybrid reward mechanism integrated with reinforcement learning (RL) training, which substantially enhances the performance of multimodal large language models across multiple visual-language reasoning benchmarks. The work also demonstrates the potential of the automated data pipeline to continually improve model performance through the ongoing incorporation of diverse data.

**Strengths:**

1. The scalable multimodal QA synthesis pipeline provides an efficient method for automatically generating domain-aware, reasoning-centric QA pairs.

2. **The WeThink dataset is a valuable new resource** with over 120K multimodal QA pairs and explicit reasoning paths, catering to diverse domains and abilities.

3. The proposed dataset have the potential to accelerate research in multimodal reasoning.

4. The case in the Appendix C offer insightful examples of the model's capabilities.

5. The paper is very well-written and easy to follow.

**Weaknesses:**

1. The paper's main strength lies in its engineering execution. However, it offers limited contributions in terms of novel, fundamental algorithms. The core methodology is largely an application or integration of existing techniques.

2. The methodology is critically dependent on the output of powerful existing models, namely DeepSeek-R1 and Qwen2.5-VL-72B, for its data generation phase. This heavy reliance makes it difficult to assess the intrinsic capabilities of the method itself, distinct from the strengths of the models it leverages.

3. The paper introduces a hybrid reward system that combines rule-based and model-based rewards. While the empirical results demonstrate its effectiveness, the design of such a composite reward function appears to be a relatively standard practice in RL for multimodal tasks, and thus offers limited novelty.

4. The claim that "our automated pipeline ensures continuous data diversity and scalable RL training" is strong. While the dataset is diverse, demonstrating *continuous* enhancement and *scalability* through empirical evidence showing performance gains with progressively larger or more diverse datasets generated by the pipeline would be more convincing. The current experiments show improvements with the dataset, but the *continuous* and *scalable* nature of the pipeline needs more direct validation.

**Questions:**

In Section 5.3, the authors note that increasing data diversity sometimes leads to performance degradation on certain benchmarks. Please provide a detailed analysis of the specific causes underlying this phenomenon, and discuss how this issue might be mitigated—for example, through improved data generation strategies or refined RL optimization approaches.

---

### Official Review · Reviewer_Mb1R · 2025-11-03

**Soundness:** 1
**Presentation:** 2
**Contribution:** 1
**Rating:** 2
**Confidence:** 4

**Summary:**

This paper proposes a method to remedy the lack of detailed and essential visual information in cold-start chain-of-thought (CoT) generated by text-only reasoning models. The approach leverages an MLLM to produce a coarse caption that is provided to a reasoning LLM, and then queries the MLLM in a multi-turn dialogue during inference to acquire additional visual details, thereby avoiding missing visual descriptions needed for reasoning. The dataset is constructed with multi-stage filtering and verification to ensure quality.

**Strengths:**

Provides a scalable data synthesis pipeline.

**Weaknesses:**

1. The motivation that cold-start CoT produced by text-only reasoning models lacks detailed, necessary visual information has already been articulated in Vision-R1 [1], which constructs pseudo-CoT to obtain richer captions and avoid this issue. The paper avoids a direct discussion of how its insight differs from Vision-R1. Moreover, the proposed data synthesis pipeline appears highly similar, e.g., both first use Qwen2.5-VL-72B to obtain captions and then DeepSeek-R1 to generate CoT trajectories. Omitting a direct comparison and discussion of the most relevant prior work is not acceptable.
2. The paper lacks direct comparisons with the most relevant approaches: it does not compare against Vision-R1, and it omits pure RL-based methods such as MM-Eureka [2], opting for selective baselines instead. I have checked the results in the works I mentioned, while  the reported results in this paper are notably lower than those of these works on math benchmarks such as MathVista and MathVerse.
3. The authors claim that raw CoT from DeepSeek-R1 is often lengthy and repetitive, making it ineffective for cold-start (e.g., Fig. 4). However, Vision-R1’s cold-start trajectories are entirely derived from DeepSeek-R1’s raw CoT and achieve performance substantially higher than reported here, which calls into question the effectiveness of the proposed method and the experimental setting.
4. The method is claimed to be scalable, but there is no validation across different data scales. Reporting results on multiple subsets of varying sizes would better substantiate the scalability claim.
5. Fig. 4 appears after Fig. 5.

[1] Vision-R1: Incentivizing Reasoning Capability in Multimodal Large Language Models. arxiv2503.

[2] MM-Eureka: Exploring the Frontiers of Multimodal Reasoning with Rule-based Reinforcement Learning. arxiv2503.

**Questions:**

N/A

---

### Official Review · Reviewer_d86K · 2025-11-07

**Soundness:** 3
**Presentation:** 3
**Contribution:** 3
**Rating:** 8
**Confidence:** 4

**Summary:**

This paper applies reinforcement learning to vision-language reasoning using a synthetic QA dataset (WeThink), generated via a multi-turn pipeline between Qwen2.5-VL and DeepSeek-R1. It introduces no fundamentally new algorithms but adapts Group-Relative PPO with a hybrid reward combining rule-based and model-based signals. The QA pairs are verified via inter-model agreement, with no human validation.

While the data pipeline is well-engineered and performance gains are consistent across benchmarks, the work falls short on methodological novelty and experimental completeness. Key issues include athe bsence of ablations on reward weighting, no analysis of RL stability, and missing comparisons to closely related baselines like Vision-R1. The empirical results suggest the dataset has utility, but broader claims of generality and robustness are not fully supported.

**Strengths:**

- question synthesis is conditioned on missing visual evidence identified by a text-reasoner, reducing premise errors common in one-shot caption→QA generation.

- system separates MC/FIB (rule-checkable) from descriptive items (model-judged), filters unverifiable questions, and requires answer agreement between a VLM and a text-reasoner, with a third model resolving conflicts concretely lowering label noise relative to single-model self-agreement.

- Raw chains are rewritten using (image, final answer) to remove redundancy and contradictions, producing shorter, more trainable rationales—an important practical step often omitted in synthetic-CoT pipelines.

- ablations show hybrid > single-source rewards and all-type RL > math-only RL, and the overall model lifts a strong Qwen2.5-VL-7B baseline across nine benchmarks

**Weaknesses:**

- “Ground truth” for QA and CoT is created solely via model filtering and cross-model agreement (DeepSeek-R1 ↔ Qwen2.5-VL with Gemini adjudication), with no reported human audit of factuality, visual grounding, or rationale fidelity leaving systematic label noise and bias unquantified.

- The hybrid objective fixes mixing weights (α_accuracy=0.7, α_format=0.3) without sensitivity or robustness analysis; the judge’s discrete tiers for descriptive items lack calibration/ablation, exposing the policy to reward hacking and miscredit.

- GRPO is presented without key controls (KL coefficient β, PPO clip ε, convergence diagnostics). Only G=5 samples per prompt are used, which yields noisy advantage estimates; yet no seed variance or training curves are reported.

- The evaluation omits head-to-head comparisons with closely related contemporaries (e.g., Vision-R1 and other RL-VL systems), making it difficult to attribute gains to the proposed pipeline rather than established recipes.

- Adding ~20k in-the-wild images improves averages but degrades MathVista/MathVerse, indicating sensitivity to data-mixture shifts and a need for curriculum or reward rebalancing to preserve domain-specific performance.

**Questions:**

- Did you run a stratified human audit of WeThink (by MC/FIB/DES and ability‐combo) to measure factuality, visual grounding, and CoT fidelity? Please report accuracy and inter-annotator agreement; if not, can you provide this in rebuttal?

- Why fix the hybrid weights (e.g., α_accuracy/α_format)? Please provide a sensitivity sweep and show how results change. For descriptive items, how is the judge calibrated (cross-judge agreement, adversarial paraphrases) to rule out reward hacking?

- What are the exact PPO clip (ε), KL coefficient (β), group size G, and seed settings? Please include learning curves and variance over ≥3 seeds to demonstrate stability with group-normalized advantages.

- What is the marginal contribution of (a) multi-turn “detail mining” vs one-shot caption→QA, (b) ability-synergy constraints, and (c) the two-stage invalid-question filter? Please quantify each on final accuracy.

- Can you add head-to-head results versus contemporaneous R1-style VL baselines under identical evaluation, and analyze the observed degradation on MathVista/MathVerse under in-the-wild scaling (mixture sweeps and/or reward reweighting)?

---

### Note · Authors · 2025-11-13

I have read and agree with the venue's withdrawal policy on behalf of myself and my co-authors.